# Cat Foster Program Outcomes: Behavior, Stress, and Cat–Human Interaction

**DOI:** 10.3390/ani12172166

**Published:** 2022-08-24

**Authors:** Kristyn R. Vitale, Delaney H. Frank, Jocelyn Conroy, Monique A. R. Udell

**Affiliations:** 1Animal Health & Behavior, Distance Education, Unity College, New Gloucester, ME 04260, USA; 2Department of Animal & Rangeland Sciences, Oregon State University, Corvallis, OR 97331, USA

**Keywords:** cat, *Felis catus*, cat behavior, cat–human interaction, social behavior, stress

## Abstract

**Simple Summary:**

Millions of cats end up in U.S. animal shelters every year. Cats living in shelters may face several stressors due to social isolation, lack of enrichment, and disturbances in their environment. Although fostering programs for dogs have been well-established in many areas, these programs are extremely rare for cats. The aim of this research was to empirically evaluate outcomes associated with placing shelter cats in a short-term foster environment, when compared with cats that remained in the shelter. Results indicate that cats placed in foster care were not at a disadvantage. Foster cats did not display decreased social behavior, increased fear or aggression, or increased cortisol levels while in the foster home. Therefore, even short-term cat fostering does not appear to be more stressful for cats than staying in the shelter. This work provides empirical evidence that cats can be placed into foster homes, even for short periods of time, when shelter space is limited.

**Abstract:**

Recent research has demonstrated that cats (*Felis catus*) have greater social potential and flexibility than was previously assumed. However, many traditional cat care practices have been influenced by the misconception that cats are socially aloof. This can result in less support or guidance for cat-focused programs that may promote improved success or welfare. For example, while dog fostering programs—even overnight programs—are considered highly beneficial, with research to back these claims, relatively little research has been dedicated to understanding the potential risks and benefits of cat fostering programs. Therefore, the aim of this study was to empirically evaluate the social, behavioral, and stress response outcomes associated with placing shelter cats in an overnight or short-term foster environment. While neither overnight nor 1-week fostering lead to a statistically significant improvement in human-directed social behavior or stress levels, foster cats also did not display increased fear or aggression in the foster home and did not have higher cortisol levels. Therefore, cat fostering—even short-term fostering—does not appear to be more stressful or problematic for this species than remaining in a shelter. This information could contribute to life-saving efforts by providing empirical evidence that cats can be safely moved into foster homes, even for short durations, when shelter space is limited. More research is needed to evaluate the potential effects of longer-term fostering in cats, as well as cat fostering practices that could lead to greater welfare benefits.

## 1. Introduction

The study of stress, behavior, and welfare among relinquished cats is a challenging topic, but one that deserves attention, given that over three million cats find themselves in shelters each year in the United States alone [1]. For both dogs and cats, living in an animal shelter has been associated with increases in physiological and psychological distress, as well as increased disease prevalence and behavior problems [2]. Among the multiple contributing factors thought to be responsible for such outcomes, social isolation is regarded as one of the major stressors for animals housed in a shelter [3]. Stress also accumulates from the exposure to high-intensity noise, strong odors, inadequate enrichment, and erratic routines [4].

An established strategy to alleviate stress in relinquished dogs involves removing the dog from the shelter and placing the animal in a temporary foster home. Fostering dogs is a growing practice with a generally positive public perception, perhaps because dogs are often thought of as highly social [5], and have been found to form beneficial bonds with human caregivers in as little as three 10-min interactions [6]. Additionally, research has shown that dogs placed in foster homes may experience decreased stress, indicated by a significant reduction in their urinary cortisol levels, compared to those that remain in the shelter [7]. Dogs living in a foster home are also more likely to use their human caregiver as a source of comfort (secure base) than dogs remaining in the shelter [8], show greater task persistence, and also score lower for neuroticism on a personality scale [9]. This trend in lowered neuroticism is likely attributed to increased opportunity for dogs in foster care to explore, socialize, and express a wider variety of their natural behaviors [10]. Even short-term fostering (one or two days) has been associated with decreased stress levels in dogs [7].

As with dog fostering programs, cat foster programs—where they exist—have been credited for significantly reducing euthanasia rates in shelters [11]. Long-term cat fostering has also been associated with increased opportunity for behavior modification and improved behavioral outcomes [11]. However, to date, there has been no research on the potential behavioral and stress benefits (or risks) of short-term (<1 week) cat foster care. In addition to potential benefits to a cat’s stress levels and behavior, short-term fostering opportunities may also be beneficial for shelters (especially during times of overcrowding, natural disasters, or other emergency events) and for potential foster caregivers who may not be able to facilitate longer foster care opportunities. Despite this, cats may not always be considered good candidates for these opportunities. For example, some foster programs offering short-term and long-term fostering for dogs only allow longer term fostering for cats: “Short Break Fostering (dogs only, not cats)—This is short-term fostering of a shelter dog, to give a dog a break from the shelter for a little while, perhaps a sleepover or a few days” [12]. The rationale for this idea is not often published. Anecdotally, concerns about the potential for increased stress and poor welfare outcomes for cats may be a primary concern. It is important to note, however, that to date, this has yet to be empirically evaluated.

Strategies that have been investigated for alleviating stress in cats residing in shelters include elements that may be readily accessible within foster homes, including access to hiding spaces, enrichment, access to additional social partners (other cats and dogs), and increased human interaction [4]. Increasing human contact in particular has shown to significantly reduce the stress response of cats [2]. Research has also demonstrated that cats are more socially flexible than originally believed [13,14]. These factors may mean that fostering cats, even for a short period of time, may be more feasible and beneficial than has previously been assumed. On the other hand, if there are certain fostering practices or durations that do lead to increased stress levels in cats or other poor outcomes, this would be equally important information in terms of establishing care practices that properly balance the potential benefits of fostering with potential risks. Therefore, the aim of this research was to empirically evaluate the social, behavioral, and stress outcomes associated with placing shelter cats in an overnight or short-term foster environment, when compared with cats that remained in the shelter with or without added socialization opportunities.

## 2. Materials and Methods

### 2.1. Subjects

Shelter cats were recruited from Willamette Humane Society (WHS) in Salem, OR, and SafeHaven Humane Society in Tangent, OR. Both shelters were no-kill, limited-admission shelters at the time of the study. Cats remained under the care of the shelter staff. Cages were cleaned daily by staff or shelter volunteers. Cage size varied based on the shelter and the number of cats in the shelter. In some cases, when space was available, cats were given access to a double-sized cage, which was two single cages without a barrier. Eighty shelter cats were initially enrolled in the research program. The demographics of the cats vary by dataset (see Appendix A). The majority of the cats were mixed breed, were spayed/neutered (except one individual who was removed after the baseline time point), and all cats were over the age of 1 year old.

Cats selected for this experiment resided exclusively in single-cat cages for the duration of the study. Cats were pseudo-randomly assigned to one of four groups, (1) control, (2) enhanced socialization within the shelter, (3) participation in a foster sleepover program, and (4) participation in a week-long foster program, with age and sex of cat considered for consistency between groups (see discussion for additional comments on group assignment). The cats in groups 1 and 2 stayed in the shelter while cats in groups 3 and 4 left the shelter to stay in a foster home. Cats in the control group received no additional socialization, while cats in the enhanced socialization group received socialization sessions with a set partner.

### 2.2. Study Time Points

Data were collected at the shelters from January 2019 until December 2020. Physiological and behavioral data were collected at three time points: (A) baseline (right before intervention), (B) 1-day during the intervention, (C) 1-day after the end of the intervention. All time points were collected at the shelter except for time point B for the foster groups, which was collected at the foster home. Urine was always collected before collecting the behavioral data.

### 2.3. Feline-ality Assessment

The Feline-ality assessment was conducted at three time points (as previously described), by a research assistant unfamiliar to the cat. Unfamiliar humans were those with which the cat had not interacted on a regular basis through socialization or regular activities at the shelter. Although some aspects of the Feline-ality assessment were standardized (e.g., the 1-min Novel Room Sociability test, as described below), other aspects of the assessment may have varied slightly among cats. For example, the time it took to complete non-timed portions of the assessment (e.g., time spent looking at the cat’s body posture in item #1) may have varied slightly between raters, therefore causing some individual difference in the time each cat spent participating in behavioral testing. Additionally, toys used in item #9, Play, varied by which toys would be available at the shelter that day. Items 1–11 of the Feline-ality assessment were followed as described within the assessment text, except for item #5, which lasted 1 min instead of 5 min. The assessments were scored using the official scale [15]. The social scores were: 1, independent; 2, social; 3, gregarious. The full Feline-ality assessment can be found on the ASPCA Pro website here: https://aspcapro.org/sites/default/files/Feline-Assessment-Forms-FINAL-05Sept13_HR.PDF (accessed on 18 August 2022).

Some cats were unable to participate in the full Feline-ality assessment due to aggressive or fearful behavior within the cage. Cats classified as fearful or aggressive displayed more than one of the following indicators: dilated pupils, lip licking, flattened ears, crouched body posture, hissing or growling, swatting/scratching, and biting. In these cases, cats were categorized as fearful/aggressive instead of given a specific Feline-ality score.

In order to assess whether participation in the intervention resulted in cats that sought to be in proximity with novel humans for a longer duration of time (something that could have adoption benefits), we examined differences in the time spent next to the unfamiliar experimenter in the first 1 min of the novel room assessment (item 5) of the Feline-ality test. The experimenter sat within a 1-m diameter circle and was inattentive to the cat (did not interact with the cat) for 1 min. The duration of time spent in proximity to the experimenter was analyzed. This measure will be referred to as “Novel Room Sociability” in the remainder of the text.

### 2.4. Urine Collection

Stress in cats is challenging to quantify. Established methods include measuring stress scores, behaviors, or hormonal responses. Cortisol is a hormone that is released by the adrenal glands in response to a stressor, and therefore has been used as a common indicator of a cat’s stress level [16]. Each collection method used to measure cortisol has advantages and disadvantages. For example, drawing blood to examine blood cortisol concentrations introduces a degree of stress due to sampling methods, but the results may be considered more accurate [16]. For fecal samples, stress is minimal for collection, but results present stress responses over a longer period of time, and post-excretion bacterial metabolism may influence cortisol measurements [17,18]. In the present study, we chose to evaluate urinary cortisol as an indicator of short-term stress, which could be collected non-invasively from the cat’s litter box (so as not to induce additional stress during collection). Urine samples reflect stress levels from a period roughly 1–3 h prior to urination. Research indicates that an increase in urinary cortisol concentration can be noted from less than 1 h after the introduction of a stressor [18] to 160 min after the stressor [19].

Urine was collected at the three time points previously described. Twenty-four hours prior to collection, each cat’s litter box was replaced with a disposable plastic litter box (17.25 × 12.34 × 3.25 inches) and the clay litter was replaced with Kit4Cat hydrophobic sand™, which is designed to repel rather than absorb urine. Urine was collected using plastic pipettes and immediately stored at 4.4 °C (40 °F), then transferred to a freezer at −15.6 °C (4 °F) for long-term storage. Samples were thawed before analysis was performed.

### 2.5. Socialization Intervention—Enhanced Socialization Group Only

Socialization sessions occurred within the shelter on three separate occasions for 15 min each (45 min total). For each session, cats were removed from their cages and brought to a small room at the shelter. Socialization partners interacted freely with the cat for 15 min before returning the cat to its cage. Interactions included petting, playing, or talking to the cat. This procedure was repeated on three separate days over the course of one week. Cats rated as fearful or aggressive were not removed from their cage, instead, the socialization partner interacted with them from the outside of their cage. The cats interacted with the same socialization partner for every session. The socialization partner was either a volunteer from the shelter or a research assistant. If the socialization partner was one of the researchers, that person would not collect behavioral data from that cat at any of the time points.

### 2.6. Fostering Intervention—Fostering Groups Only

Neither shelter had an adult cat foster program already in place prior to the study. The only foster programs that were in place previously were for kittens or adult cats with health issues. In all, 15 newly recruited cat foster caregivers hosted cats from the study, with seven taking more than one foster cat (at different time periods). One foster took eight different foster cats throughout the duration of the study.

Cats stayed at the foster home for either 1 night or 1 week. Volunteer fosters would pick up and drop off their foster cats at set times. At the pick-up time, fosters met with a researcher to receive a bag of supplies. These included a paper Feline-ality assessment, a written survey to track activities in the foster home, food, and litter supplies. The survey contained questions about the foster cat, these included: Were they confined to one room or did they have full access to house? What type of activities did you engage in with your foster cat? Roughly how much time did you spend interacting with your foster cat during the program? Litter supplies included a clean, non-disposable litter box and regular clay litter (both used and provided by the shelter), and the supplies to collect urine (disposable litter box, hydrophobic litter, pipette/tube for collection). If it was a foster’s first time taking a cat, the instructions for Feline-ality and urine collection were explained. Fosters were told that the disposable litter box (which all cats had also used at baseline) should be set out on the evening of the first day of fostering and left out until a sample was received, or once 24 h had passed, and the Feline-ality assessment should be performed on the first full day of foster, following collection of urine. After explaining this, the cat would be sent home with the foster. Cats were removed from their cage by a researcher, placed in a cat carrier, and transported to the house by the volunteer foster.

### 2.7. Data Analysis

Cortisol Analysis: Urine samples were run in three separate batches, with an equivalent number of individuals from each treatment group included within each batch. For all batches, Immulite 1000 Siemens (Healthcare Diagnostics) instruments were used to run the samples. The Immulite 1000 uses a solid-phase, chemiluminescent immunoassay in a bead-contained unit to assay cortisol concentration. The first batch was run by the experimenters. For this batch, each sample had two results, and the average of these results was taken for analysis. The other two batches were run by the Oregon Veterinary Diagnostic Laboratory. Because urinary cortisol can be affected by water intake, the cortisol:creatinine ratio was calculated to standardize the cortisol level. Creatinine was analyzed by the Oregon Veterinary Diagnostic Laboratory. All samples were run on an AU480 Beckman Coulter, which measures creatinine using a modified Jaffe procedure. Cortisol was then standardized using the following equation:Urine cortisol (μg/dL) × 0.0276/Urine creatinine (mg/dL) × 88.40 = Urinary Cortisol Creatinine Ratio (UCCR)

Behavior Analysis: As is typical, Feline-ality scores were calculated during the assessment by the experimenter. Items 1–11 of the Feline-ality sheet were scored as the assessment was being run. Using the official scale, cats were assigned social scores of independent (low sociability); social (medium sociability); or gregarious (extremely social).

Novel Room Sociability videos were coded by a primary coder using the behavioral data collection application *Countee*. The proportion of time spent in proximity to the person (within the circle) was analyzed. In order to assess inter-observer reliability (IOR), 30% of the videos were double coded by a secondary coder using *Countee*. Scores for the two videos were considered “in agreement” with one another if they were within 8% of each other.

### 2.8. Statistical Analysis

Data were analyzed using non-parametric statistics. Two-tailed Wilcoxon signed-rank tests were used for repeated measures, and Mann–Whitney U tests were used for comparisons between groups; these tests were run using the Social Science Statistics Calculator. Individual data were examined using a Fisher’s exact test run in GraphPad. All statistical tests had an alpha level of 0.05.

## 3. Results

### 3.1. Fostering Activities

As mentioned, fosters completed a written survey to track activities with the cat during their stay in the foster home. Our results indicate that foster cats were either confined to one room during their stay at the home (10 cats), had full access to the home (6 cats), or had partial access to the home (10 cats). Additionally, fosters reported engaging in several different activities with their foster cat (Figure 1). Of these, playing was the most frequently reported activity foster volunteers engaged in with their foster cat (20 responses). Petting was also commonly reported (16 responses), as well as cuddling/snuggling (13 responses), both of these activities can be seen in Figure 2a. Although not captured in the reported activity survey, some people also socialized their foster cats to novel conspecifics, including cat-friendly dogs (Figure 2b). On average, volunteers who took cats for 1 night reported spending an average of 7.6 h with the cat. Volunteers who took cats for 1 week reported spending an average of 17.95 h in total with the cat.

### 3.2. Urinary Behavior and Cortisol Analysis

At baseline, 72 cats urinated in the hydrophobic litter. Eight cats did not urinate in the litter box at baseline and were excluded from further analysis. An additional 10 cats were removed from the study after baseline (e.g., cat was adopted, removed due to medical reasons, moved to a multi-cat cage, etc.). A total of 61 cats had urinary data at both time points A and B, and a total of 49 cats had urinary data at both time points A and C (additional cats were removed from the study after time point B due to the above reasons).

Fisher’s 2 × 2 exact tests were run to examine the number of foster cats that experienced an increase or a decrease in cortisol from time point A to B, and from time point A to C. Only cats with numerical urinary cortisol creatinine ratio (UCCR) scores were included in this analysis. When comparing 1-day foster cats to 7-day foster cats, no significant difference was seen in the number of cats that had an increase or decrease in UCCR scores from A to B (Table 1, Fisher’s 2 × 2, *p* = 0.061) or from A to C (Table 1, Fisher’s 2 × 2, *p* = 0.37).

Fisher’s 2 × 2 exact tests were also run to examine the number of cats staying in the shelter that exhibited an increase or a decrease in cortisol from time point A to B and from time point A to C. When comparing cats that received in-shelter socialization to control cats, no significant difference was seen in the number of cats that had an increase or decrease in UCCR scores from A to B (Table 1, Fisher’s 2 × 2, *p* = 0.4283) or from A to C (Table 1, Fisher’s 2 × 2, *p* = 1).

Because of the small sample sizes in the four groups, data were combined into two groups for additional statistical analysis. One group included the cats going to overnight or 7-day foster (i.e., foster group) and the other group included the cats that remained in the shelter, with or without socialization opportunities (i.e., shelter group). There was also no significant difference between the foster and shelter groups in the number of cats displaying an increase or a decrease in cortisol from time point A to B (Fisher’s 2 × 2, *p* = 0.28) or from time point A to C (Fisher’s 2 × 2, *p* = 0.20).

One unexpected result was identified within the first day of the intervention (time point B). As mentioned, only cats that urinated in the litter box at baseline (A) continued to participate in the study. Despite all foster cats urinating at baseline while in the shelter, several foster cats did not urinate in the litter box at time point B, which was collected at the foster’s home. The number of foster cats that did not urinate at B was compared to those cats that stayed in the shelter. Significantly fewer foster cats urinated in the litter box on day 1 of the intervention (B) compared to cats staying in the shelter (Table 2, Fisher’s 2 × 2, *p* = 0.027). However, after returning to the shelter at time point C, no significant difference was found in the number of cats using the litter box in the foster or in-shelter groups (Table 3, Fisher’s 2 × 2, *p* = 0.49).

For cortisol analysis, intra-assay reliability was calculated for the first batch of cortisol samples run by the experimenters. The four controls used to assess the reliability of the cortisol measurements showed cortisol levels of 21.7, 22, 23, and 22.9 μg/dL. The average was 22.4 μg/dL with a standard deviation of 0.648 μg/dL. The intra-assay CV for these samples was 2.983.

In order to analyze cortisol levels between groups before the intervention to levels during and after the intervention, the UCCR was compared between time points for each group. Only cats that had a numerical UCCR score were included in this analysis. When comparing all foster cats at baseline, before the intervention, to 1 day during the intervention (A vs. B), no significant difference in UCCR score was seen (Wilcoxon signed-rank test: *N* = 11, Time Point A Score, *M* = 0.0000043, *SD* = 0.000011, Time Point B Score, *M* = 0.0000064, *SD* = 0.0000058, *W* = 18, *p* = 0.18). The same result was found when comparing baseline to 1 day following intervention (A vs. C), and no significant difference in foster cat UCCR scores were detected (Wilcoxon signed-rank test: *N* = 19, Time Point A Score, *M* = 0.0000046, *SD* = 0.0000082, Time Point B Score, *M* = 0.0000046, *SD* = 0.0000034, *W* = 69, *p* = 0.29).

Similarly, no significant difference in UCCR scores were seen for cats staying in the shelter when comparing time points A and B (Wilcoxon signed-rank test: *N* = 25, Time Point A Score, *M* = 0.0000052, *SD* = 0.0000039, Time Point B Score, *M* = 0.0000048, *SD* = 0.0000030, *W* = 106.5, *p* = 0.13) or at time points A and C (Wilcoxon signed-rank test: *N* = 21, Time Point A Score, *M* = 0.0000047, *SD* = 0.0000035, Time Point B Score, *M* = 0.0000043, *SD* = 0.0000019, *W* = 75, *p* = 0.16).

To compare between groups, the UCCR scores of the foster group were compared to the scores of cats staying in the shelter. For cats with data at A and B, no significant difference was found between groups at baseline (Time point A, Mann–Whitney *U*(36) = 94, *Z* = 1.48, *p* = 0.139) or on the first day of intervention (Time point B, Mann–Whitney *U*(36) = 116, *Z* = -0.72, *p* = 0.47). The same results were found for cats with data at A and C, again no significant difference was found between groups at baseline or 1 day following intervention (Mann–Whitney *U*(40), both *p* > 0.05).

In order to examine whether cortisol levels differed between the two shelters, UCCR was compared between the shelters at each time point. Only cats that had numerical UCCR scores were included in this analysis. When comparing between the two shelters, no significant difference in UCCR score was noted at any of the time points (Time point A, Mann–Whitney *U*(66) = 399, *Z* = -1.84, *p* = 0.066; Time point B, Mann–Whitney *U*(39) = 154.5, *Z* = 0.91, *p* = 0.36; Time point C, Mann–Whitney *U*(42) = 201.5, *Z* = 0.36, *p* = 0.72).

Finally, in order to examine whether time spent at the shelter impacts cortisol levels (as seen in dogs [20]), the times spent at the shelter before baseline UCCR measurement were compared. For cats with a numerical UCCR score and an intake date, the number of days since intake date until baseline test were compared to the UCCR score. A strong correlation was not seen between UCCR score at baseline and the number of days the cat had been in the shelter (*R*^2^ = 0.071).

### 3.3. Social Behavior

Of the 72 cats that urinated at baseline, 71 participated in the Feline-ality assessment at baseline. As mentioned previously, additional cats were removed from the study after baseline (e.g., cat was adopted, removed due to medical reasons, moved to a multi-cat cage, etc.). A total of 59 cats had Feline-ality data at both time points A and B, and a total of 54 cats had Feline-ality data at both time points A and C.

The social score of the Feline-ality test (numerical rating on the scale of independent to social to gregarious) were examined. Cats labeled as aggressive/fearful were not included in these analyses as they did not have a numerical rating. Fisher’s 2 × 2 exact tests were run to examine the number of foster cats that experienced an increase or a decrease in the social score of the Feline-ality assessment from time point A to B and from A to C. When comparing 1-day foster cats to 7-day foster cats, no significant difference was seen in the number of cats that had an increase or decrease in social scores from A to B (Table 4, Fisher’s 2 × 2, *p* = 0.20) or from A to C (Table 4, Fisher’s 2 × 2, *p* = 1).

Fisher’s 2 × 2 exacts tests were also run to examine the number of cats staying in the shelter that had an increase or a decrease in social score of the Feline-ality assessment from A to B and from time point A to C. When comparing cats that received socialization to control cats, no significant difference was seen in the number of cats that had an increase or decrease in social scores from A to B (Table 4, Fisher’s 2 × 2, *p* = 0.59) or from A to C (Table 4, Fisher’s 2 × 2, *p* = 0.58).

To determine whether foster cats displayed more social behavior in a home environment than in the shelter environment, social score data were combined into two groups for additional statistical analysis (i.e., foster group vs. shelter group). There was also no significant difference between the foster and shelter groups in the number of cats displaying an increase or a decrease in social score from time point A to B (*N* = 50, Fisher’s 2 × 2, *p* = 0.32) or from time point A to C (*N* = 39, Fisher’s 2 × 2, *p* = 0.082).

To examine problem behaviors in the home compared to the shelter, the number of cats displaying signals of aggression/fear during the Feline-ality test was examined. No significant difference was found for the foster group before their fostering experience compared to during the foster experience (Table 5, Fisher’s 2 × 2, *p* = 1).

When comparing before the fostering experience to 1 day after they returned to the shelter, no significant difference was found in the number of foster cats displaying aggression/fear (Table 6, Fisher’s 2 × 2, *p* = 0.35).

Additionally, no significant difference was found in the number of aggressive/fearful cats when comparing foster cats to cats that stayed in the shelter at time points A (Fisher’s 2 × 2, *p* = 0.11), B (Fisher’s 2 × 2, *p* = 0.06), or C (Fisher’s 2 × 2, *p* = 0.21). However, it should be noted that for time point B, there was a trend toward significance, with seven cats that stayed in the shelter displaying aggressive/fearful behavior compared to only one foster cat. Overall, cats did not display a significant change in fear or problem behavior in the home environment compared to the shelter environment.

Comparisons were also conducted for overall Feline-ality type, with the number of foster cats ranked as highly social (gregarious types, which include Love Bug, Personal Assistant, Leader of Band) compared to the number of cats in the less-social Feline-ality types (independent and social). Out of 25 foster cats, 9 were labeled as gregarious in the shelter at baseline and 12 were labeled as gregarious during their stay in the foster home, which was not a significant difference in the number of highly social cats (Fisher’s 2 × 2, *p* = 0.57).

The two shelters were compared in terms of both cats displaying aggressive/fearful behavior and social Feline-ality types. The number of cats displaying signals of aggression or fear during the Feline-ality test were examined between Shelter 1 (S1) and Shelter 2 (S2). There was no significant difference between the shelters in terms of cats displaying aggressive/fearful behavior at baseline (S1, *N* = 34, Agg/Fear = 4; S2 *N* = 37, Agg/Fear = 5; Fisher’s 2 × 2, *p* = 1), at time point B (S1 and S2, *N* = 30, Agg/Fear = 4; Fisher’s 2 × 2, *p* = 1), or at time point C (S1, *N* = 26, Agg/Fear = 5; S2 *N* = 29, Agg/Fear = 9; Fisher’s 2 × 2, *p* = 0.37). The number of cats ranked as highly social (gregarious types) were also compared between shelters at each time point. Again, there was no significant difference between the shelters in terms of number of cats ranked as highly social at baseline (S1, *N* = 34, Social = 6; S2 *N* = 37, Social = 7; Fisher’s 2 × 2, *p* = 1), at time point B (S1, *N* = 30, Social = 12; S2 *N* = 30, Social = 10; Fisher’s 2 × 2, *p* = 0.79), or at time point C (S1, *N* = 26, Social = 5; S2 *N* = 29, Social = 8; Fisher’s 2 × 2, *p* = 0.54).

To determine whether foster cats showed greater sociability toward an unfamiliar person in the post-test compared to pre-test, the results from the Novel Room Sociability data were analyzed. An inter-observer reliability score of 83% was found for the Novel Room Sociability data (agreement for 19/23 videos). No significant difference in sociability was noted for foster cats from time point A to C (Wilcoxon signed-rank test: *N* = 21, Time Point A Score, *M* = 0.18, *SD* = 0.35, Time Point C Score, *M* = 0.12, *SD* = 0.35, *W* = 61.5, *p* = 0.48). Novel Room Sociability scores were also compared between the foster and shelter groups. No statistically significant differences between these groups were identified at either time point A or C (Mann–Whitney, all *p* > 0.05).

## 4. Discussion

Our results indicate that although short-term fostering did not lead to a statistically significant improvement in human-directed social behavior or stress levels, foster cats also did not display more aggression or fear in the foster home, did not display a decrease in social behavior, and did not display an increase in UCCR after returning to the shelter when sent to foster for 1 week or even just 1 day. Therefore, cat fostering—even short-term fostering—may not be more stressful or problematic for this species than remaining in a shelter and could contribute to life-saving efforts by allowing cats to move into homes when shelter space is limited. More research is needed to evaluate the potential effects of longer-term fostering in cats, as well as cat fostering practices that could lead to greater welfare benefits.

The finding that foster cats did not urinate during the first night of foster care may indicate that lack of urination is an alternative behavioral marker of stress [17] or a behavioral side effect of temporary relocation. This is a question that warrants future exploration. However, after returning to the shelter, cats resumed using the litter box, and for 1-week fosters that did not urinate in the litter box on the first night of foster, none were reported to have issues urinating in the box on day 2 of foster. This suggests that this effect is not long lasting when it occurs. Given the significant number of cats displaying this behavior, it is likely that this could also happen to a meaningful number of cats during a rehoming event, which suggests that both foster providers and new owners might benefit from being informed that for many cats, litter box aversion may be expected for up to a day into the transition period.

Although cats may have experienced some stress due to the change in environment, as noted by the litter box aversion while at foster, they still freely participated in social interactions, as measured by the Feline-ality data. Cats were equally likely to be highly social both at the shelter and at foster. Additionally, social interactions such as playing, petting, and cuddling/snuggling, were the most frequently reported activities foster volunteers engaged in with their foster cats (Figure 1). Prior work indicates that shelter dogs and cats seek out social interactions, and even spend more time with an inattentive stranger than pet dogs and cats [21,22]. This may be due to differences in opportunities for social interactions. In comparison to pet cats that have a stable caretaker that can provide frequent social interaction, shelter cats may live in a more deprived social environment. Given that social isolation can be a major stressor for shelter animals [3], that human interaction can be preferred even over other appetitive stimuli (e.g., food, toys, or scents) [23], and that human interaction can be a beneficial form of enrichment for shelter cats [4], this makes opportunities for social interaction extremely important. In all, cats in the foster environment appear to receive a benefit through the opportunity to engage in social interactions and receive individualized attention.

For cats staying within the shelter, no significant differences were found between cats in the control group and cats in the enhanced socialization program in terms of cortisol score or Feline-ality social score (Table 1 and Table 4). One possible explanation for this is that participating in behavioral testing itself may have been enriching for the cats staying within the shelter [24]. In both the control and socialization groups, cats spent around 10–12 min engaging in the Feline-ality assessment at each time point (with exact times varying between raters, as described in Section 2.3). During the Feline-ality assessment, cats received attention, petting, and play. These activities may have caused cats in both in-shelter groups to engage in more social interaction over time. This is supported by the data, the majority of in-shelter cats displayed an increase in social score from A to B and from A to C, rather than a decreasing score (Table 4). It is also possible that there is an upper limit to how much socialization is beneficial to a cat. Cats in the socialization group received an additional 45 min of socialization with a set partner on top of the Feline-ality assessment interactions. Although longer socialization sessions can produce more affiliative social behavior toward humans (when comparing kitten handling sessions of 15 to 40 min), another study found that compared to kittens handled 1 h a day, kittens handled 5 h a day were not friendlier toward people [25]. Therefore, for adult cats, lengthening the amount of socialization past a certain amount may not produce a beneficial effect. The exact amount of socialization needed to produce a beneficial effect for shelter cats should be the topic of future research.

There are several challenges and limitations to consider for this study, such as the number and breed distribution of subjects. Within the shelters, we were working with the number of cats available during the study period, which was unpredictable, and the proportion of cats eligible for foster was inconsistent. In part, this was due to the rapid rate of adoptions during the study period, which, while beneficial for the cats, created some challenges in terms of collecting a full dataset for a larger number of cats. Some cats were also prevented from going to foster, or from being placed on temporary hold to fully participate in the study, because they were breeds deemed to be highly adoptable. This biased our sample toward mixed-breed cats. However, given the applied nature of this work, these challenges are representative of conditions that would impact shelter/foster programing more generally. For example, cats prevented from going to foster by the shelter for this study would likely also not be sent to foster outside of the study. In addition, the results obtained with a moderate sample size are useful for informing decisions about fostering at the individual or single-shelter population level, especially given our use of within-subject measures, and provide a foundation for future studies with larger cat populations. Without this initial evidence that short-term fostering in cats does not appear to produce a long-lasting increase in stress levels or significant changes in aggressive or fearful behavior, it may have been difficult to conduct a larger scale study on this topic ethically.

Overall, the data suggest that there may not be significant reasons to avoid placing cats in foster care, even overnight or in short-term care, due to behavioral or stress-related concerns, especially given that there are documented benefits associated with fostering cats [11], as well as providing enrichment through human interaction [4]. Additionally, in our study, three different foster volunteers ended up adopting their foster cat. This indicates that short-term fostering may also be a viable option for improving adoption outcomes. Future research could examine how fostering programs could be used as a means to allow potential adopters time to ensure their home is a good fit for the cat, and how this impacts the retention of the cat within the home and long-term welfare outcomes. In all, shelter workers and fosterers were open to the idea of adult cat foster opportunities and were excited to be involved in research that had the potential to increase the welfare of shelter cats.

## 5. Conclusions

Some may believe cats to be socially aloof, and that cat care practices that involve short-term fostering opportunities place shelter cats at a disadvantage. However, the results of the current study indicate that shelter cats placed in a foster home, even for just 1 day, do not display an increase in stress (as measured by UCCR and fearful/aggressive behavior) or a decrease in social behavior toward humans. The finding that foster cats did tend to display litter box aversion on the first night of foster is an important consideration. Foster volunteers, as well as cat caretakers bringing their newly adopted shelter cat home, should be informed that this may be a temporary behavior related to relocation. Prior research indicated that social interaction is an important aspect of shelter cats’ lives. Shelter cats seek out human social interaction [22], and even tend to prefer human social interaction over other appetitive stimuli [23]. Because of this, providing opportunities for human social interaction is extremely important for shelter cats. Foster cats can benefit from the chance to engage in social interactions with a set caregiver and receive individualized attention. In all, this is important information for animal shelters, which are often extremely limited on time, space, and resources. Our findings indicate that shelter cats can be placed in foster homes, even for short periods of time, when additional space within the shelter is needed. Future research should examine the impact of long-term fostering on cats (>1 week), as well as additional fostering practices that may lead to beneficial outcomes for shelter cats.

## Figures and Tables

**Figure 1 animals-12-02166-f001:**
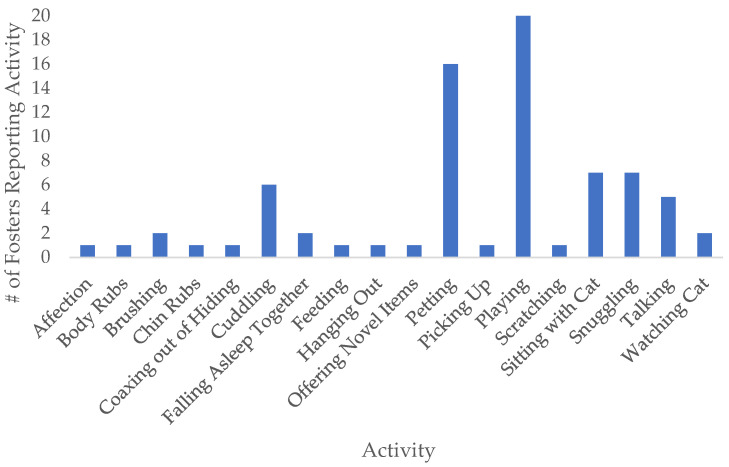
Results of the written survey to track which activities volunteers reported engaging in with their foster cats during their stay at the foster home.

**Figure 2 animals-12-02166-f002:**
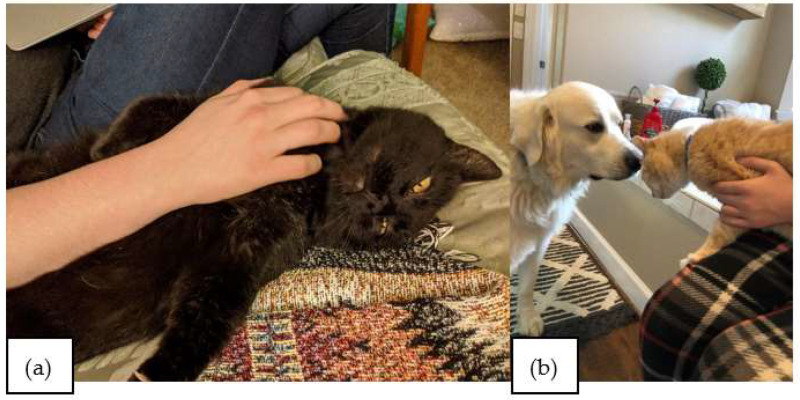
Photos of cats while at the foster home. (**a**) Displays a cat sitting next to their foster while receiving petting. (**b**) Displays a foster cat being introduced to a cat-friendly dog, the foster cat responds by rubbing their head.

**Table 1 animals-12-02166-t001:** Number of cats in each foster group and each in-shelter group that had an increase or decrease in UCCR from time point A to B and A to C.

Time Points	Group	Increase	Decrease	Total
	1-Day Foster	6	0	6
A–B	7-Day Foster	2	3	5
	1-Day Foster	7	3	10
A–C	7-Day Foster	4	5	9
	Socialization	8	6	14
A–B	Control	4	7	11
	Socialization	4	8	12
A–C	Control	3	6	9

**Table 2 animals-12-02166-t002:** Number of cats urinating in the litter box at time point B by group.

Group	Did Urinate in Box at B	Did Not Urinate in Box at B	Total
Foster	19	10	29
In-Shelter	29	3	32

**Table 3 animals-12-02166-t003:** Number of cats urinating in the litter box at time point C by group.

Group	Did Urinate in Box at C	Did Not Urinate in Box at C	Total
Foster	22	0	22
In-Shelter	25	2	27

**Table 4 animals-12-02166-t004:** Number of cats in each foster group and in-shelter group that had an increase or decrease in the Feline-ality social score from time point A to B and A to C.

Time Points	Group	Increase	Decrease	Total
	1-Day Foster	7	6	13
A–B	7-Day Foster	10	2	12
	1-Day Foster	7	5	12
A–C	7-Day Foster	4	4	8
	Socialization	10	3	13
A–B	Control	11	1	12
	Socialization	9	1	10
A–C	Control	7	2	9

**Table 5 animals-12-02166-t005:** Number of foster cats displaying aggressive/fearful behavior at time points A and B.

Time Point	No AGG/Fear	Yes AGG/Fear	Total
A	26	1	27
B	26	1	27

**Table 6 animals-12-02166-t006:** Number of foster cats displaying aggressive/fearful behavior at time points A and C.

Time Point	No AGG/Fear	Yes AGG/Fear	Total
A	24	1	25
C	21	4	25

## Data Availability

All data supporting reported results can be found in the Appendix A.

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
