# Peer review of "Cat Foster Program Outcomes: Behavior, Stress, and Cat–Human Interaction"

_animals, 2022, doi:10.3390/ani12172166_

Round 1
Reviewer 1 Report
This paper presents interesting and novel data comparing behavior and cortisol levels of cats in foster care (for either one night or one week) and those remaining at the shelter (either with or without an enhanced socialization program). Overall, the data suggest that fostering is not stressful for cats based on behavioral measures and cortisol levels, with the caveat that many cats did not urinate in the litter box during the first day in foster. No differences were found for cats with and without the enhanced socialization program. The sample sizes are okay, given the challenges of conducting research with shelter animals (e.g., adoption during the study and the need for research to conform to shelter policies). My specific comments are detailed below:
Abstract
Line 28: I found the phrase “short-term fostering (< 1 week)” a little confusing – at first reading, I was not sure this included the overnight group as well as the one week group. Perhaps replace with “While neither overnight nor one-week fostering lead to a statistically...”
Introduction
Line 51: I am not sure that I would describe fostering dogs as a “common” shelter practice (do we have that data?), although “growing” seems accurate.
Materials and Methods
I might have missed it, but what was the time period for the study?
Lines 98-102: I would like to know more about the two shelters (e.g., no-kill? open admission? size of cat populations? average length of stay for cats, if available) and the care and housing of the cats at each (e.g., cage type and dimensions; type and frequency of enrichment; how long are cats out of their cages each day/week).The only information currently provided are the names of the two shelters and that cats are singly housed. Also, a brief description of the transport of the cats between shelter and foster homes is needed.
Regarding cat characteristics, did you record the time each cat had been at the shelter when they entered your study? In dogs, cortisol levels change with time at shelter (e.g., Hennessy et al. 2001. Applied Animal Behaviour Science 73:217-233), so this would seem to be an important variable to measure in cats as well. It would be helpful to include descriptive statistics for time at shelter for each of your four groups to make sure they did not differ in this regard.
Can you provide the Felinality Assessment as an Appendix? Some readers might not be familiar with it.
Section 2.6 Fostering intervention: did you provide the foster homes with advice or information about fostering? If so, please describe it. How did you get the reports of what activities occurred in the foster home (e.g., phone calls, emails, written surveys)?
Section 2.8 Statistical Analysis: Did you look at whether the behavioral results and cortisol levels differed between cats at the two shelters?
Results
Lines 222-225: How many cats had urinary cortisol measurements for all three time points (A,B,C) and did you look for patterns in these data?
Lines 249-258: For the first day (measured at time point B), do you know whether the cats held their urine or urinated outside of the litterbox?
Discussion:
Lines 380-383: The description (especially counts) of activities engaged in by volunteers and their foster cats should also be included in the Results section.
The Discussion focuses exclusively on fostering and does not cover results from the two shelter groups, with and without an enhanced socialization program. The fact that no differences in cortisol or behavior were found between these two groups should be discussed in the context of results from other efforts at enrichment for shelter cats. Any ideas as to why you found no differences?
Author Response
Thank you for your feedback. Please see the attachment.

Reviewer 2 Report
Dear Authors, I am pleased to review your manuscript. I am glad that the problem of the welfare of shelter cats is being discussed more and more often by scientists. In my opinion, your study and the results obtained are a good pilot study for other scientists working to improve the welfare of shelter cats. I have posted my comments below.
line 110-111 Was it the same person for all cats for the entire duration of the study? did each cat have a different person? Part of the answer to my question was found on line 165.
line 113-114 what was the total experimental time for each cat? Was it the same for everyone?
2.4. Urine collection, In my opinion, it is worth supplementing this fragment (or adding a fragment in the introduction) with more information about cortisol in the urine. How long after a stressor has been triggered may a change in urine cortisol levels appear? (line 150: how much short-term stress?)
line 249-258 Here I have a few questions about the litter box itself. Did the cat go to the foster house with its litter box from the shelter? And with the same litter in the litter box? If not, was the litter box in the temporary home the same type as the one he used in the shelter? Similarly, how was the litter? I agree with you that the lack of urination in the initial period of the stay at the foster house could have been caused by stress. However, cats can also have very strong preferences for the litter box and litter.
The doubts that arose to me while reading the manuscript were mostly dispelled in the further parts of the manuscript. My review is ultimately shorter than it was at the beginning because I found the answers to my questions in the further parts of the text.
In my opinion, the work requires only minor corrections and more precisely supplementing some content. As for those already presented, I have no objections.
Author Response

(The authors gave the same response as above.)

Round 2
Reviewer 1 Report
Thank you for addressing all of my comments from my previous review. In this second review, for the question "Are the results clearly presented?" I chose "can be improved" because there are problems in the numbering of tables. In the revision, Table 2 was deleted (information merged with Table 1) and Table 6 was deleted (information merged with Table 5), but numbers of other tables were not adjusted; there are six tables in the revision, not eight. All table numbers after 1 should be checked on the table itself as well as in the text of the Results and Discussion sections where tables are called out.
Author Response
Reviewer Comment: Thank you for addressing all of my comments from my previous review. In this second review, for the question "Are the results clearly presented?" I chose "can be improved" because there are problems in the numbering of tables. In the revision, Table 2 was deleted (information merged with Table 1) and Table 6 was deleted (information merged with Table 5), but numbers of other tables were not adjusted; there are six tables in the revision, not eight. All table numbers after 1 should be checked on the table itself as well as in the text of the Results and Discussion sections where tables are called out.
Author Response: Thank you very much for pointing out this oversight. The numbers of the tables in the table captions and text of the paper have been updated. Thank you again for your helpful feedback!